

# Advancing opinion mining with optimised explicit feature extraction in customer reviews

Rajeswary Santhiran[1], Kasturi Dewi Varathan[1] and Yin Kia Chiam[2]

[1] Department of Information Systems, Faculty of Computer Science & Information Technology, Universiti Malaya, Kuala Lumpur, Malaysia
[2] Department of Software Engineering, Faculty of Computer Science & Information Technology, Universiti Malaya, Kuala Lumpur, Malaysia

## ABSTRACT

The growing emphasis on opinion mining highlights the significance of analysing customer opinions regarding a service or product and their impact on purchasing decisions. Although identifying pertinent features within customer review analysis is essential for uncovering the expectations, extracting features from unstructured customer review documents presents significant challenges. Current pattern rules are also insufficient for extracting relevant explicit features, alongside linguistic limitations hindering the processing of customer review documents. Thus, this study enhanced the performance of explicit feature extraction from customer reviews by improving heuristic pattern-based rules. The rules comprised 16 newly constructed rules and 25 rules derived from previous studies. Notably, these 16 new rules could extract explicit features frequently overlooked by existing rules in past studies. An enhanced heuristics pattern-based algorithm was also created to identify and extract explicit features using a set of 41 enhanced heuristic pattern-based rules. Although fewer rules were employed for explicit feature extraction compared with previous studies, no optimisation was used for either the rules or the feature extraction process. Consequently, an average precision of 0.93, a recall of 0.90, and an F-measure of 0.91 were obtained across seven datasets from multiple domains. This proposed algorithm exhibited consistent performance and adaptability across various domains, underscoring the effectiveness of the proposed enhanced heuristics pattern-based approach. Overall, the outcomes demonstrated improved explicit feature extraction capabilities, enabling precise identification of specific product attributes or services referenced by customers. This advancement could then facilitate a more profound understanding of customer preferences, pain points, and desires, with significant practical implications for businesses aiming to comprehend and address their customers' needs.

Corresponding author
Kasturi Dewi Varathan,
kasturi@um.edu.my

# INTRODUCTION

A significant surge in the expansion of online businesses concerning electronic commerce has been observed in the past decade. One recent publication (*Intelligence Node, 2024*) reported that 24% of shoppers engaged in product research focused on quality and deals. Conversely, 23.5% read online customer reviews. Hence, the influence of online activities on product sales is becoming increasingly challenging to dispute. Examples of these activities include marketing and e-reviews.

The Local Consumer Review Survey 2024 (*Padget, 2024*) conducted by BrightLocal revealed that 91% of customers in the United States (US) acknowledged the impact of reviews on their overall perception of brands. In contrast, 77% of customers utilised at least two review platforms before making a business decision. This survey then demonstrated that 59% of customers expected that a business should possess between 29 and 99 reviews before considering its services. Furthermore, 96% of customers discovered review search functionality beneficial, while 69% expressed increased confidence in using a service after reading positively written reviews. The survey also noted that a single negative review could deter approximately 30 potential customers. Therefore, the increasing significance of online reviews in influencing customer purchasing decisions is evident.

*Mumuni et al. (2020)* uncovered that the importance of product reviews substantially influenced the value provided to customers. The study stated that customers regarded a review as relevant when it evaluated the characteristics (or features) of a product or service of interest. Similarly, *Pooja & Upadhyaya (2024)* highlighted the significant relationship between review credibility and content that provided information regarding products or services. The study presented that customer reviews were highly dependent on their characteristics. Examples of these characteristics included pricing, longevity, and customer support.

*Siddiqui et al. (2021)* and *Ran et al. (2021)* asserted that the utilitarian functions and content of reviews were vital for establishing credibility among customers. The study denoted that customers primarily sought extensive information concerning product characteristics when reading reviews. Examples of this information were specific features, their advantages and disadvantages, the benefits offered, and the experiences of others related to these features. Thus, feature extraction from customer reviews has been demonstrated as an essential component of all these studies.

Features are initially identified using noun terms, in which a term that frequently appears in a document is regarded as a feature (*Pak & Günal, 2022*). Frequency also serves as a criterion for feature selection, with terms failing to meet a specified threshold being excluded (*Liu, 2022*). Nonetheless, frequency alone or the occurrence of noun terms in a document does not inherently qualify as a feature. The base assumption that features are exclusively nouns or noun phrases has proven inadequate, as it overlooks certain identified features that are non-nouns (*Mahmood, Abbas & ur Rehman, 2023*). Despite the selection of all nouns from the documents, the performance of feature extraction does not reach a superior outcome (*Mahmood, Abbas & ur Rehman, 2023*).

Many studies have utilised noun for feature extraction. *Pak & Günal (2022)* and *Tran, Duangsuwan & Wettayaprasit (2021)* employed nouns or noun phrases to extract features.

*Park & Kim (2022)* then identified non-noun terms as features. Another earlier study by *Maharani, Widyantoro & Khodra (2015)* reported that one of the datasets in their research contained over 100 non-noun features. Meanwhile, *Asghar et al. (2019)* and *Tubishat, Idris & Abushariah (2021)* introduced verb patterns to extract features that were previously overlooked by noun-based rules. Conversely, all these methodologies encompass only a limited set of rules. This observation suggests that additional patterns beyond nouns and adjectives should be explored for feature and opinion extractions. The necessity is due to a limited number of rules governing the variation of feature or opinion types, resulting in the omission of many features (*Tubishat, Idris & Abushariah, 2021*).

The feature extraction performance is significantly affected by the presence of irrelevant features. *Prastyo, Ardiyanto & Hidayat (2020)* emphasised the importance of improving the extraction algorithm. The study noted that an improved algorithm could effectively handle feature space reduction, retrieve and retain only pertinent features, and increase learning accuracy. More recently, the precision value reported by *Mahmood, Abbas & ur Rehman (2023)* was only 0.85 based on a rules-based algorithm for feature extraction. Therefore, establishing a defined feature extraction process is vital for improving algorithm performance.

This study focused exclusively on features paired with opinion words to enhance the explicit feature extraction from customer product review documents. The outcomes from this study could achieve its objective through three primary contributions as follows:

i) The introduction of a novel set of pattern rules for feature extraction could address a limitation of previous rules

ii) The creation of an enhanced heuristic pattern-based algorithm for explicit feature extraction

iii) The implementation of a domain-independent algorithm applicable across different datasets

This study is structured as follows: The Related Works section discusses existing studies, while the Proposed Methodology section details the framework and methodology employed. The Results and Discussion section then present the experimental work conducted on publicly available datasets. Finally, the Conclusions and Future Works section summarises the findings of this study.

## RELATED WORKS

Customer reviews are closely related to satisfaction, reflecting opinions or feedback regarding the disparity between expectations and actual experiences (*Laksono et al., 2019*). This phenomenon can affect the purchasing decisions of other consumers. One market research study performed in the US (Local Consumer Review Survey 2023; *Padget, 2023*) revealed that 98% of customers engaged with online reviews, with 70% expressing trust in these reviews. The study also demonstrated that a single negative review could dissuade up to 30% of prospective customers. Thus, customer reviews, while rooted in personal experience, can serve as a potent marketing tool to enhance product or service promotion. Simultaneously, these reviews are capable of significantly undermining sales.

Many studies have increasingly focused on feature extraction, improving performance through existing methods, proposing entirely new approaches, or employing a hybrid of multiple methods. These methods are usually categorised into three types according to their dependence on training data: (i) supervised, (ii) semi-supervised, and (iii) unsupervised. Supervised methods necessitate training data for model training, whereas unsupervised methods operate without any training data (*Yu, Gong & Xia, 2021*). These processes also aim to discover all relevant features that are subject to opinion.

*Rasappan et al. (2024)* proposed an optimised machine learning (ML) algorithm known as the enhanced golden jackal optimiser-based long short-term memory (EGJO-LTSM) algorithm. This algorithm could calculate optimal weights in sentiment analysis of customer reviews. The study utilised optimisation techniques of EGJO for classification and a combination of improved grey wolf optimiser (IGWO) and log-term frequency-based modified inverse class frequency (LF-MICF) for term weighing and feature selection. This approach also incorporated components from particle swarm optimization (PSO) and an adaptive weighted strategy to minimise feature size for feature selection. Consequently, the precision was improved up to 25% compared to other state-of-the-art methods.

*Kaur & Sharma (2023)* proposed a hybrid methodology that combined Term Frequency-Inverse Document Frequency (TF-IDF), co-occurrence frequencies, and long short-term memory (LTSM) approaches to extract features pertinent to reviews and aspects for classification purposes. The hybrid feature vector-LTSM technique involving different methods then demonstrated enhanced performance relative to other leading strategies utilising the same dataset for feature extraction and classification. Thus, the outcomes of the study effectively addressed the challenges associated with accurately identifying features from reviews for summarisation.

*Mahmood, Abbas & ur Rehman (2023)* introduced a hybrid method that utilised the type dependency relations (TDR) method from dependency relation analysis and pattern rules to extract product aspects and opinions from Amazon reviews concerning cell phones and food. The study also pruned the extracted aspects using a product list to reduce the number of features from the extraction. Moreover, SentiWordNet was employed to assess polarity further. Consequently, the proposed method demonstrated superior performance compared to baseline studies, achieving a precision of 85% and a recall of 75%.

*Bhuvaneshwari et al. (2022)* documented a Bi-LSTM self attention based convolutional neural network (BAC) model for subjectivity classification. This model could address challenges concerning unstructured customer review sentences. Examples of these issues included lengthy sentences, thwarted expectations, and logical complexities in polarity detection. The model was also designed to utilise pre-trained word embeddings, in which identified features were input into different BAC layers to build a semantic relationship. Superior performance was then accomplished by the model, attaining higher reliability compared to standalone models referenced in the study. On the contrary, the model reported higher computational time.

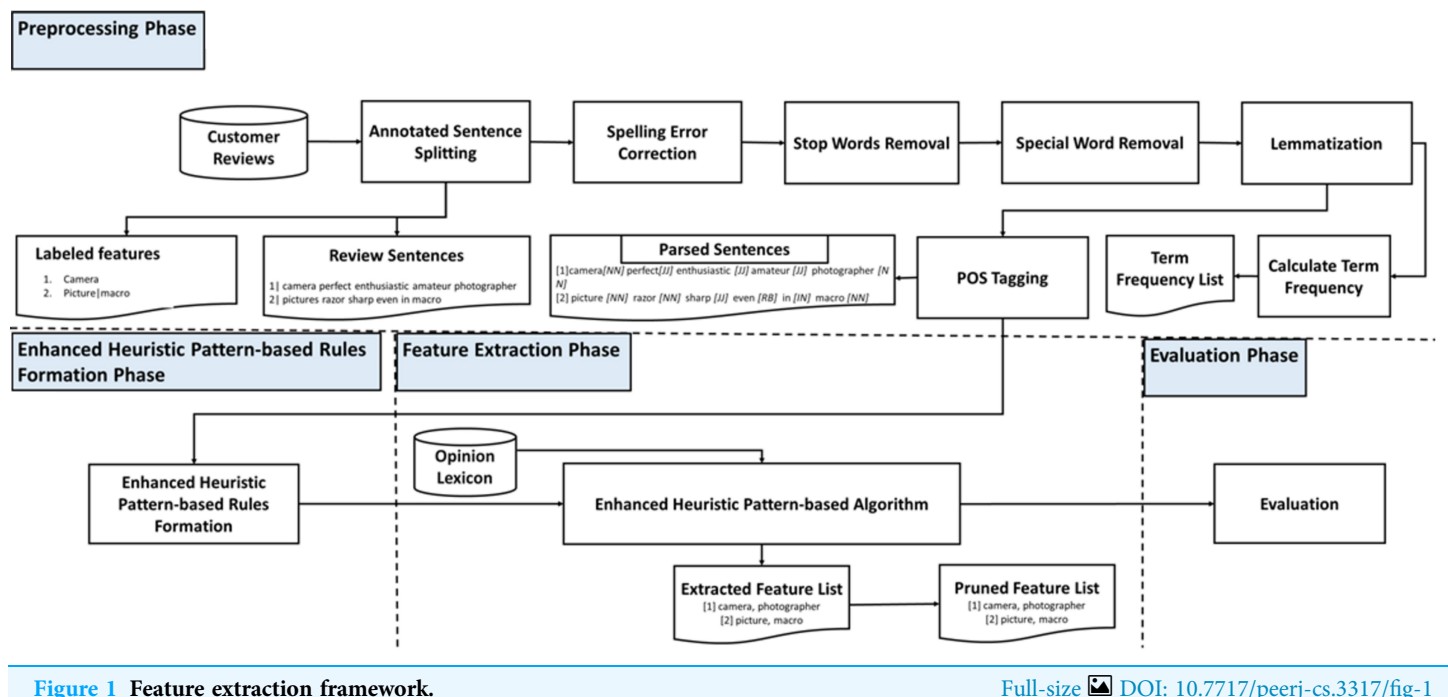

**Figure 1** Feature extraction framework.

# RESEARCH METHODOLOGY

This study presented a novel framework for explicit feature extraction through an enhanced heuristic pattern-based algorithm applied to customer reviews. Figure 1 depicts the proposed framework, comprising four primary phases: (i) preprocessing, (ii) enhanced heuristic pattern-based rules formation, (iii) feature extraction, and (iv) evaluation. The following subsections provide a detailed description of each phase.

## Dataset

Publicly available real-world datasets extracted from customer review portals were employed for academic research. Table 1 tabulates further descriptive details regarding these datasets.

## Tools and resources

The experiments conducted in this study were developed and executed using Python, incorporating four standard Python libraries: (i) NLTK, (ii) NumPy, (iii) Pandas, and (iv) CSV. A Jupiter Notebook also served as a development tool on Windows 10 Pro, whereas model and data libraries from the Flair framework were employed for part-of-speech (POS) tagging (*Akbik et al., 2019*).

This study used two established opinion lexicons created by *Hu & Liu (2004)* and *Almatarneh & Gamallo (2018)*, which were publicly accessible. The opinion lexicon (https://www.cs.uic.edu/~liub/FBS/sentiment-analysis.html) constructed by *Hu & Liu (2004)* received significant citations from researchers in sentiment analysis and opinion mining. This lexicon was manually curated and contained 6,800 positive and negative

**Table 1 Dataset description.**

| No | Reference | Datasets | Domain | Dataset size |
|---|---|---|---|---|
| 1 | *Toledo-Ronen et al. (2022)* (https://aclanthology.org/2022.naacl-main.198/) | Restaurants, Electronics, Hotels, Automotive, Movies, and others | Multi-domain | 671 |
| 2 | *Jiang et al. (2019)* (https://aclanthology.org/D19-1654/) | Food | Restaurant | 501 |
| 3 | Sem Eval 2016 (https://aclanthology.org/S16-1002/) | Restaurant | Restaurant | 1,430 |
| 4 | Sem Eval 2016 (https://aclanthology.org/S16-1002/) | Laptop | Computing | 500 |
| 5 | Sem Eval 2014 (https://aclanthology.org/S14-2004/) | Restaurant | Restaurant | 200 |
| 6 | Sem Eval 2014 (https://aclanthology.org/S14-2004/) | Laptop | Computing | 500 |
| 7 | Amazon Product Reviews (2004) (https://www.cs.uic.edu/~liub/FBS/sentiment-analysis.html#datasets) | Digital cameras, cell phones, MP3 players, DVD player | Electronic | 1,679 |

sentiment words. Likewise, the opinion lexicon (https://github.com/citiususc/VERY-NEG-and-VERY-POS-Lexicons) by *Almatarneh & Gamallo (2018)* included a compilation of extreme opinion words. This lexicon comprised 57,207 words. The experiment codes (https://github.com/rajeswary3/feature-extraction) developed for this study were also publicly available to promote reusability, ensuring transparency and reproducibility of this research.

## Preprocessing phase

Customers' reviews often articulate preferences, opinions, prior expectations, and emotions regarding the purchase and use of a product or service. These reviews are often written in native languages and contain grammatical and spelling errors. Thus, text cleaning is critical for maintaining a high-quality process. The procedure encompasses removing spaces and special characters, stop words removal, and stemming or lemmatization. This study then employed WordNetLemmatizer (https://www.nltk.org/api/nltk.stem.wordnet.html) and stopwords (https://www.nltk.org/api/nltk.corpus.html) from the NLTK library for lemmatization and stop word removal, respectively. Meanwhile, part-of-speech tagging was executed utilising the Flair tagger (https://flairnlp.github.io/docs/tutorial-basics/part-of-speech-tagging) library in Python.

## Enhanced heuristic pattern-based rules formation phase

A final heuristic pattern-based rule for explicit feature extraction was developed using two sets of rules. Table 2 presents the initial set, which comprises 25 rules extracted from past studies. This study then constructed novel heuristic pattern-based rules for feature extraction that were absent from the initial set of rules. Subsequently, additional rules were formulated to extract more precise features, aiming to enhance the accuracy of explicit feature extraction in this study.

Despite evidence indicating that certain features included non-noun words (such as verbs), previous studies predominantly focused on noun-based words as features. This outcome indicated that features could be derived from non-noun words, and opinions were not limited to adjectives alone. Hence, this study developed new rules to address the oversight of specific features that did not belong to the common noun category. Even though

**Table 2 Feature extraction pattern rule from existing studies.**

| No | First word | Second word | Third word | Study |
|---|---|---|---|---|
| 1 | AVB | JJ | NN/NNS | Asghar et al. (2019) |
| 2 | NN | NN | NN/NN | Maharani, Widyantoro & Khodra (2015) |
| 3 | JJ | NN/NNS | | Asghar et al. (2019), Maharani, Widyantoro & Khodra (2015) |
| 4 | JJ | NN | NN | Asghar et al. (2019) |
| 5 | JJ | JJ | NOT NN/NNS | Maharani, Widyantoro & Khodra (2015) |
| 6 | JJ | TO | VB | Asghar et al. (2019) |
| 7 | JJ | VB/VBN/VBD | NN/NNS | Maharani, Widyantoro & Khodra (2015) |
| 8 | NN | TO | NN/NNS–NN/NNS | Maharani, Widyantoro & Khodra (2015) |
| 9 | NN | [RB] | NN/VB[O] | Maharani, Widyantoro & Khodra (2015) |
| 10 | NN | IN | NN | Asghar et al. (2019), Maharani, Widyantoro & Khodra (2015) |
| 11 | NN | JJ | | Asghar et al. (2019) |
| 12 | NN | NN/NNS | JJ | Asghar et al. (2019), Tubishat, Idris & Abushariah (2021) |
| 13 | NN | [IN+DT]+NN+[VBP] | JJ [O] | Tubishat, Idris & Abushariah (2021) |
| 14 | NN | VBZ-RB | JJ [O] | Tubishat, Idris & Abushariah (2021) |
| 15 | NN | VBZ | JJ [O] | Tubishat, Idris & Abushariah (2021) |
| 16 | NN | VBZ+RB+JJ[O] | NN | Asghar et al. (2019) |
| 17 | NN/NNS | IN | DT–NN/NNS | Asghar et al. (2019) |
| 18 | NN/NNS | JJ | NOT NN/NNS | Tubishat, Idris & Abushariah (2021) |
| 19 | NN/NNS | JJ | | Asghar et al. (2019) |
| 20 | NN/NNS/RB/RBR/RBS | JJ/VBN/VBD | | Asghar et al. (2019) |
| 21 | PRP | VB | DT+NN | Asghar et al. (2019) |
| 22 | RB/RBR/RBS | JJ | NN/NNS | Htay & Lynn (2013) |
| 23 | VB | NN/NNS | | Asghar et al. (2019) |
| 24 | VB | JJ | | Asghar et al. (2019) |
| 25 | VB | VB | NN | Asghar et al. (2019) |

**Note:**
NN/NNS–Singular/Plural Nouns, JJ–Adjectives, AVB/RB/RBR/RBS–Different types of adverbs, VB/VBZ/VBD–Different types of verbs, IN–Prepositions, DT–Determiner, O–Opinions, A–Features.

features and opinions are commonly associated with nouns and adjectives, this new rule set identified distinct patterns for features and nouns. Table 3 details the newly established rules formulated based on an analysis of sample datasets and observational techniques.

## Feature extraction phase

The tokenised and parsed review sentences from the preprocessing stage were aligned with the pattern rules to extract opinionated features. These extracted features then underwent pruning before the generation of the final list. As a result, this study proposed a novel heuristic pattern-based algorithm for explicit feature extraction to accomplish the above functionality.

### Enhanced heuristics pattern-based algorithm

Optimal practices in the design and development of a model are critical for improving execution performance, resulting in time and computational resource savings. Therefore, the framework in this study adopted a modular design, segmenting the entire model into

**Table 3 Newly formed feature extraction pattern rules.**

| No | First word | Second word | Third word | Example |
|---|---|---|---|---|
| 1 | JJ(A) | VB(O) | | *Infrared blessing* |
| 2 | NN | NN(O) | | *Vibration top* |
| 3 | RB(O) | NN | | *Better **speakerphone** ever* |
| 4 | JJ(O) | JJ | | *Incredibly crappy **remote**; pretty **sturdy*** |
| 5 | VB | JJ | | ***Screen** great; **read** seconds* |
| 6 | NN | RB(O) | | *Treat **battery** well last* |
| 7 | JJ(O) | VB/JJ | NN | *Outstanding signal **reception*** |
| 8 | JJ | RB | NN/NNS | *Shooting scene tough automatically **focus*** |
| 9 | NN | RB+VBP | NN [O] | **Case** ever make quality |
| 10 | NN | RB[O] | NN | **Controls** especially scroll wheel |
| 11 | NN | RB | NN/JJ [O] | **Backlit screen** infinitely better |
| 12 | NN/NNS | VB/IN/NN/NNS | NN/NNS [O] | **Software** getting favorite |
| 13 | NN/NNS | VB/IN/NN/NNS | NN/NNS/JJ [O] | **Scroll wheel** select push straight |
| 14 | VB [A] | IN-DT/IN | JJ [O] | **Read** within seconds |
| 15 | JJ [O] | NN-VB | NN/NNS | Best bet looking **phone** |
| 16 | NN | VB-RB | JJ [O] | **Speaker** makes even great |

**Note:**
NN/NNS–Singular/Plural Nouns, JJ–Adjectives, AVB/RB/RBR/RBS–Different types of adverbs, VB/VBZ/VBD–Different types of verbs, IN–Prepositions, DT–Determiner, O–Opinions, A–Features. Word in bold are features extracted based on the given rule pattern.

several components. These components could then function autonomously and were designed for repeated parallel use, facilitating scalability and diverse executions with multiple datasets. The independent components could also be modified to utilise an alternative set of pattern rules without impacting the core logic and processing. Figure 2 illustrates the enhanced heuristics pattern-based algorithm in extracting explicit features from preprocessed review sentences that adhere to the specified pattern rules. These rules were designed in two or three words, necessitating a thorough examination of the sentence in bigram or trigram modes to verify the fitting pattern of the token (word).

The developed iteration or looping functionality could minimise unnecessary memory consumption. This loop sequence was intended to commence with a review sentence, after which the pattern group would validate the rules before proceeding to the following review sentence to avoid unnecessary iterations. One prominent example was that all rules were validated before processing the subsequent sentence upon reading a review sentence. This validation of the rules occurred in a grouped sequence. One example was the group of rule patterns initiated with nouns, which was validated first. This process was followed by the group that began with verbs. This is explained in Fig. 3. Nonetheless, reversing the sequence could cause the loop to iterate over the dataset size for each rule. This process then necessitated a substantial amount of time and resources. Effective data structure use was also considered for storing review sentences and initialising local variables. One significant example was employing dictionaries from the Python library. Concurrently, using publicly accessible pre-existing libraries could substantially enhance performance of the algorithm.

**Algorithm : Heuristic Pattern-based Feature Extraction Algorithm based on rules and opinion lexicon**

**Input:**
sI      = POS-tagged pre-processed sentence
rL      = Heuristic pattern-based rule-list
oL      = Opinion Lexicon
dS      = term frequency document
tH      = Frequence threshold (set as 2)
resNER = NER (Name Entity Recognition) that is not accepted as Feature

**Output:**
fF      = Final Feature List

**Output:**
Extracted Feature List
Rule_Opinion_Feature List

```
    Begin
1   for each sentence in sI do:
     begin
2      while not (rL..eof)
        begin
3        for each token in sentence do
          begin
4           for each rule in rL.
             begin
             // to tokenize each pattern exist in a rule and repeat for all patterns in the rule
5             for each pattern in the rule do:
               begin
                 // to compare word token matches rule pattern
6                if (token.tag matches pattern)then :
                  begin
7                   read token.next.tag or read token.previous.tag
                    // to extract previous word and next word if currect word token matches the current pattern
8                   if (token.tag matches rule.next_pattern) or (token.tag matches rule.previous_pattern) do:
9                    if (token.position matches rule.feature_position)
                      begin
                       // indentifying the positions of feature and opinions if matching patterns in rule
10                     read token.position as featPos
11                     read token.word at token.position(featPos) as wrd_feat
12                     read rule.opinion_position as OpPos
13                     read token.word at token.position(OpPos) as wrd_Op
                       // 1st step of pruning- pruning based on opinion
14                     if (wrd_Op exist in oL) and (wrd_feat not in oL) do:
                        // add the feature-opinion set into list
15                       add wrd_feat and sentenceId in featurelistArray
16                       add rule-wrd_Op-wrd_Feat combo and sentenceId in featOpinionListArray
                         exit
                        end if
                      end if
                    end if
                  end if
                end for
              end for
           end while
         end for

    // Feature pruning for the extracted features
17  read dS
    //2-step Pruning: based on frequency threshold
18  for each features in fL do:
     Begin
19    check dS.frequency(feature) as featFreq
      // add features that fullfil the threshold into final feature list
20    if (featFreq more or equal to tH) do:
21     add feature in fF List
      end if
    end for

    //3-step Pruning: based on NER words
22  for each features in fF do:
     Begin
23    check NER for feature
24    if (feature.NER is part of resNER) do:
       // remove feature from final feature list
25     remove feature from fF
      end if
    end for
    end
```

**Figure 2** **Heuristic pattern based feature extraction algorithm.**

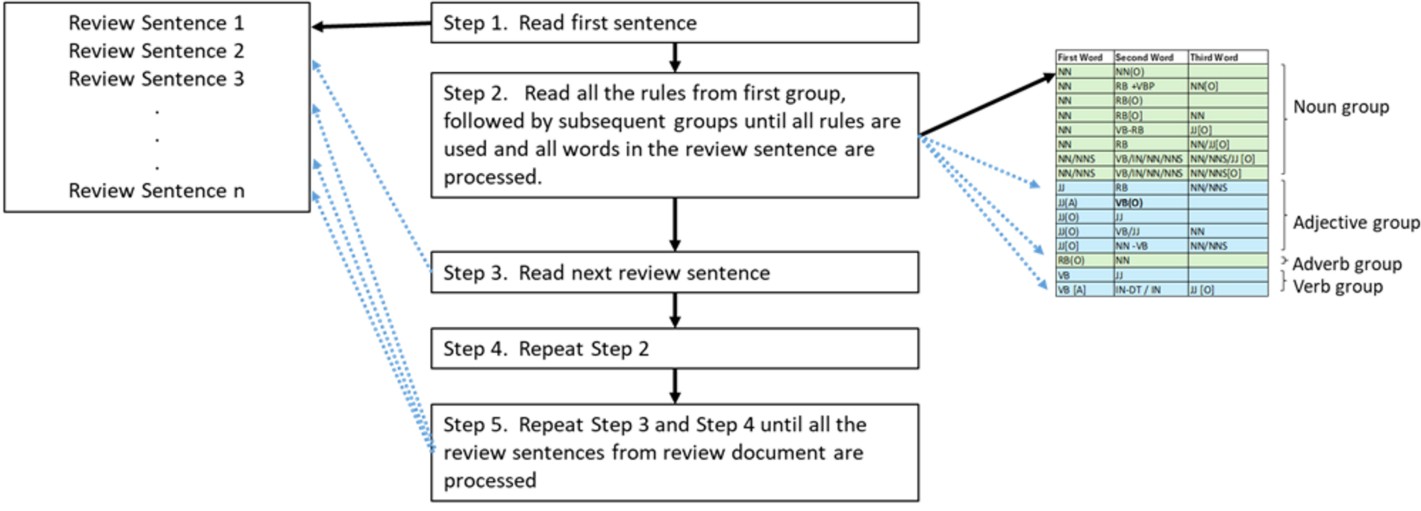

**Figure 3 Review sentence processing using rules.**

The final segment of this phase included feature pruning. This process could reduce the number of the extracted features, preserving only the relevant ones. The extracted features then underwent a three-stage pruning process before the algorithm produced a final list of extracted features. Initially, features lacking opinion word associations were pruned immediately during the extraction process. In this context, opinion lexicons were employed to ascertain whether the terms associated with the features were opinion words. The second stage of pruning then affirmed the frequency of the extracted feature by using the threshold and frequency list generated during preprocessing. Finally, the last stage of feature pruning involved named entity recognition (NER), which validated the entity of the feature based on NER labels to ensure its relevance as an attribute of the product or service. Figure 4 displays the NER labels used to categorise relevant and irrelevant features for feature pruning in this study. The final list of extracted explicit features that successfully passed all pruning stages was produced as the outcome of the enhanced heuristic pattern-based algorithm. The process of pruning is explained below:

Example:

Review sentence after preprocessing:

*nokia/NN try/VB very/RB hard/JJ again/RB pretty/RB good/JJ csr/NN*

"nokia" will be extracted as a feature as it fulfills the below pattern

Pattern : NN–VB-RB-JJ [nokia/NN try/VB very/RB hard/JJ]

"nokia" will go through all the pruning stages as depicted in Fig. 5 and gets pruned when it fulfills one of the three pruning criteria.

### Feature pruning

The execution of the feature extraction algorithm does not typically guarantee that the extracted features are relevant to the subject or can improve performance (*Bhamare & Prabhu, 2021*). Irrelevant features are likely being selected alongside the significant ones. Thus, the inclusion of all extracted features without assessing their

NER Labels

**Labels for Relevant Feature**

- PRODUCT
- EVENT
- LOC
- QUANTITY
- CARDINAL
- (NO LABEL)

**Labels for Feature Pruning (Irrelevant Features)**

- PERSON
- NORP
- ORG
- GPE
- WORK OF ART
- LANGUAGE
- DATE
- TIME
- PERCENT
- MONEY
- ORDINAL

**Figure 4 NER labels for features.**

Feature pruning based on opinion words

Feature pruning based on frequency

Feature pruning based on NER label

Extracted Feature - "nokia" is accompanied by opinion word ="hard"

Hence, **Feature –nokia will not be pruned** under this Condition

```
phone|60
nokia|169
small|15
size|20
proprietary|1
headset|1
color|90
screen|40
piece ]1
good|67
powerful|1
```
Term Frequency List

Based on Term Frequency List above, extracted Feature-nokia is having a frequency above the threshold value ( threshold value =2)

Hence, **Feature- nokia will not be pruned** under this condition

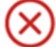 nokia ORG

NER label "ORG" is part of NER labels ("ORG" represents organization) for Feature Pruning (refer to Figure 3)

Hence, **Feature- nokia will be pruned** under this condition

**Figure 5 Three-stage pruning.**

significance can introduce extraneous features, increase feature size, and decrease extraction performance (*Xie et al., 2023*). This observation implies that improvement of the extracted features list is essential before the evaluation phase. Hence, this study implemented feature pruning to reduce the size of the extracted features.

*Feature pruning based on opinion*

Only features associated with an opinion must be extracted in a feature extraction process aimed at identifying opinionated features. Conversely, certain scenarios are also observed where a feature may meet the heuristic pattern rule without an associated opinion. In the event of such a scenario, the identified feature should be pruned. Meanwhile, an opinion word can be cross-referenced with the opinion lexicon to confirm its status as a recognised opinion word when it is identified as associated with a feature based on the pattern rule. Consequently, a feature is pruned if it is identified without an opinion word or if the extracted opinion word is not recognised in the opinion lexicon. The following samples are some identified examples:

> Review sentence: "*Although a guide is provided, the configuration is easy.*
> Preprocessed review sentence and observed patterns:
> *Although\IN guide\NN provided\VB the\DT configuration\VB easy\VB*

i) (see Table 2) Rule #20: "guide/NN provided/VB," The extracted feature is "guide," but the associated word "provided" does not exist in the opinion lexicon. Therefore, the extraction process will eliminate the "guide."

ii) (see Table 2) Rule #11: "configuration/NN easy/JJ," where "configuration" is the extracted feature and is accompanied by the opinion word "easy." Hence, "configuration" will be retained in the extracted feature list.

This study presented 16 new patterns that demonstrated opinion words could belong to any part of speech, which was contrary to previous studies that primarily focused on adjectives (*Ahmad, Shaikh & Tanwani, 2023*; *Pak & Günal, 2022*; *Rana et al., 2021*). Hence, limiting opinion words to adjectives could constrain the expression of feedback or customer perspectives. Certain examples included words listed in the Opinion Lexicon (https://github.com/almatarneh/LEXICONS) that did not function as adjectives, including "thoroughly", "relaxing", "timely", and "uplifting". The fundamental principle applied here pertained to semantic relevance, wherein the feature under review coexisted with an opinion word within the sentence. Alternatively, the feature was perceived as irrelevant in the absence of an association with an opinion word (*Tubishat, Idris & Abushariah, 2021*; *Zhang, Wang & Liu, 2019*).

*Feature pruning based on frequency*

*Hu & Liu (2004)* conducted a pioneering study that proposed frequency as a pruning evaluation criterion for the elimination of infrequently occurring features based on a specific threshold. Although the extracted attribute is generally identified as a feature and linked to an associated opinion word, its occurrence in the review documents categorised it as an outlier. These low-frequency features can then enhance the feature list. Pruning

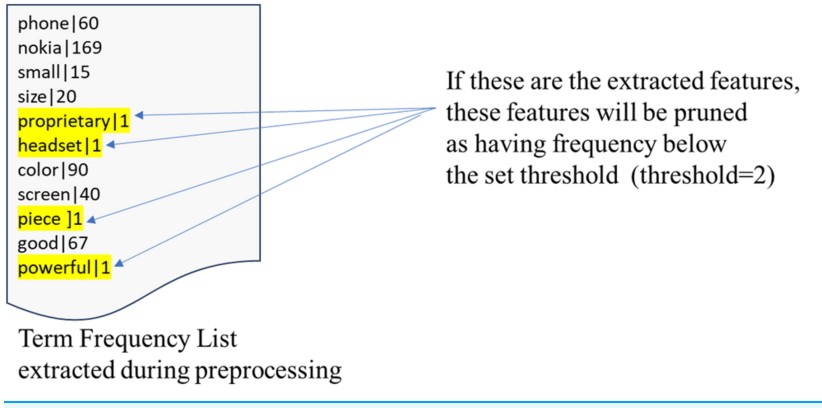

**Figure 6  Term frequency list.**               

certain features can also improve performance and result quality. Numerous studies have then adopted frequency-based pruning, demonstrating its effectiveness in eliminating less essential features of products or services (*Akhtar, Garg & Ekbal, 2020*; *Chauhan & Meena, 2020*; *Tubishat, Idris & Abushariah, 2021*). The frequency with which the feature appears in the reviews indicates its significance to customers. Hence, applying a frequency threshold for feature pruning can remove features of lesser significance. This process can optimise the extracted features while improving extraction performance.

This study employed the term frequency list generated by the preprocessing algorithm for frequency pruning during the review *corpus* processing. The term frequency list consisted of the count of occurrences for each word presented in the document. Thus, this algorithm could prune the extracted features based on the threshold. The threshold value was determined using the elbow method (*Al-Tamimi & Shkoukani, 2023*). Based on example given in Fig. 6 which depicts the term frequency list, it can be seen that the extracted features "proprietary", "headset", "piece" and "powerful" appear only once throughout the review document. Since the frequency is below the threshold, the features will be pruned.

*Feature pruning based on NER*
This study presented a novel approach to employing NER for feature pruning. Numerous entities associated with a product or service (brand or location) could be referenced in customer reviews. In contrast, these entities did not represent actual characteristics of the product or service. Instead, the entities were considered supplementary to the feature list. The application of NER to identify product name entities was also substantiated by recent research findings (*Bharadi, 2022*; *Lee & Kim, 2023*). This NER was supported by the SpaCY language model, recognising 18 entities. Examples of these entities included individuals, nationalities, facilities, organisations, countries, locations, products, events, artworks, legal documents, and language.

The feature pruning algorithm eliminated features identified in NER labels, excluding those related to individuals, organisations, events, and product labels. This NER is extensively utilised for extraction in deep learning applications. Nonetheless, this study

aimed to identify the attributes of the reviewed entity rather than to extract the entity itself. Given that NER could identify entity names rather than common attributes, pruning the NER label in features could help in eliminating entities. This process could then lead to decreased number of the extracted features. These labels were selected based on features observed in the training data. Features that were not among the selected labels for pruning were not pruned. An example is then given as follows:

Review sentence: "*csr[+1]##nokia is trying very hard, and again – they do have pretty good csr's.*"

After preprocessing

Review sentence: nokia/NN try/VB very/RB hard/JJ again/RB pretty/RB good/JJ csr/NN

Labelled/Annotated feature: csr

Extracted Features:

Feature #1: "nokia "(Rule pattern: NN–JJ)

Pattern: NN–VB-RB-JJ [nokia/NN try/VB very/RB hard/JJ]

Feature #2: "csr" (Rule pattern: JJ–NN)

Pattern: JJ–NN [good/JJ csr/NN]

NER label for both the features as follows:

Nokia: ORG

csr: CARDINAL

NER label "ORG" which represents an organisation is part of NER labels for feature pruning for this study (see Fig. 4)

Hence, Feature#1 will be pruned under this condition

NER label "CARDINAL" is not part of NER labels for Feature Pruning for this study (see Fig. 4)

Hence, Feature#2 will not be pruned under this condition

This example implied that Feature#1 was extracted as it fulfilled one of the pattern rules. Nevertheless, Feature#1 was deemed irrelevant, as the labelled feature for this review sentence was csr (Feature #2, which was also extracted by one of the established rules). Hence, Feature#1 was removed through feature pruning.

## Evaluation phase

### Evaluation metrics

The evaluation criteria were derived from *Liu et al. (2016)*, who calculated precision, recall, and F-measure utilising TP (true positive), TN (true negative), FP (false positive), and FN (false negative) values. Precision typically refers to the proportion of accurately identified features relative to the total identified features. Conversely, recall indicates the percentage of identified features compared to the total labelled features.

## RESULTS AND DISCUSSION

Considering that evaluation is paramount for validating the proposed framework and its components, this section provides and analyses the assessment results comprehensively.

**Table 4 Explicit feature extraction performance.**

| Amazon product review dataset | Enhanced patten-based algorithm using 25 rules | | | Enhanced patten-based algorithm using 41 enhanced rules | | |
|---|---|---|---|---|---|---|
| | Precision | Recall | F-measure | Precision | Recall | F-measure |
| Nokia | 0.9 | 0.87 | 0.89 | 0.94 | 0.91 | 0.92 |
| Creative lab | 0.84 | 0.81 | 0.83 | 0.91 | 0.88 | 0.90 |
| Canon | 0.88 | 0.85 | 0.87 | 0.94 | 0.91 | 0.92 |
| Nikon | 0.82 | 0.8 | 0.81 | 0.92 | 0.92 | 0.92 |
| Apex | 0.8 | 0.77 | 0.79 | 0.82 | 0.8 | 0.81 |
| **Average** | **0.85** | **0.82** | **0.84** | **0.91** | **0.88** | **0.89** |

**Note:**
The average results of the datasets are shown in bold.

## Performance evaluation of the identified rules in extracting explicit features

Two experiments were conducted utilising 25 pattern rules and 41 enhanced pattern rules. Notably, the 41 enhanced rules comprised 16 newly constructed rules and 25 from previous studies. Table 4 lists the average performance of the five electronic datasets. The 41 enhanced rules indicated increases in precision, recall, and F-measure values of 7.06%, 7.32%, and 5.95%, respectively. This outcome suggested that the inclusion of 16 additional rules facilitated the identification of features previously neglected in earlier studies. Simultaneously, the explicit feature extraction performance across all five electronic review datasets was substantially enhanced. Therefore, these novel rules effectively identified pertinent and accurate features, in which the 16 additional rules constituted a significant artefact that enhanced the extraction of explicit features. The process then led to improved performance, reflecting the capacity of the model to achieve a balance between accuracy and completeness in identifying relevant and actual features from the datasets. Furthermore, the model exhibited strong reliability in accurately identifying explicit features articulated through opinions.

## Performance evaluation of feature pruning impact on explicit feature extraction

Figure 7 illustrates the precision performance of feature extraction both with and without feature pruning. A three-stage pruning was implemented on all five Amazon product view test datasets: (i) pruning based on opinion association, (ii) pruning based on frequency thresholding, and (iii) pruning based on NER. As a result, the pruning of extracted features through these stages eliminated noise and irrelevant ones, improving the quality of the retained features and the extraction performance. The feature extraction precision then improved significantly, averaging an increase of 3.4% across all five datasets.

## Performance evaluation of enhanced heuristic pattern-based algorithm for explicit feature extraction

The performance of the proposed model was evaluated by comparing its explicit feature extraction performance capabilities with benchmark studies utilising the same Amazon
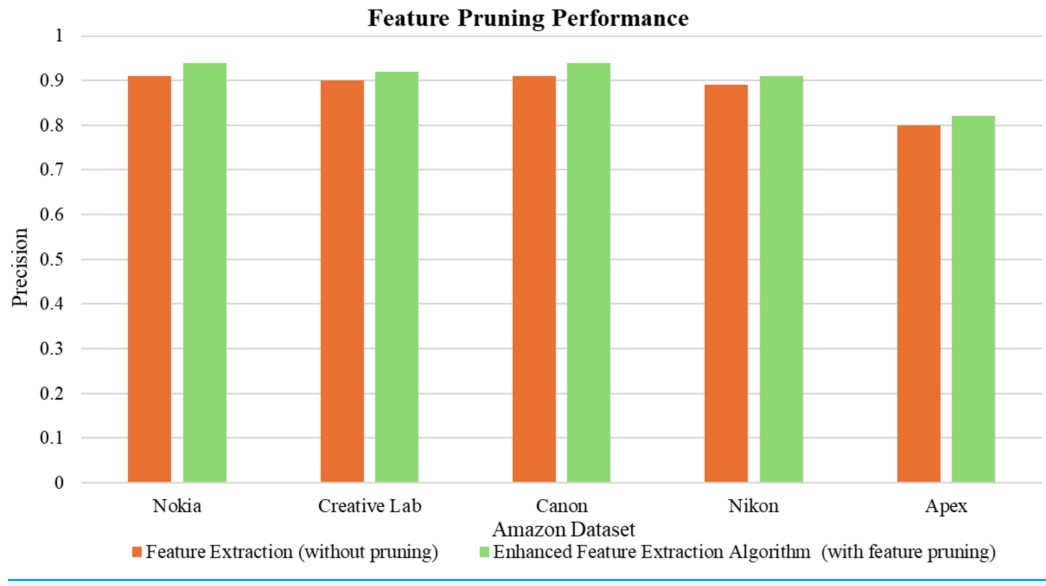

**Figure 7 Feature pruning performance.**

Product Review datasets. Table 5 provides a comparison of performance indicators (precision, recall, and F-measure) and the number of rules used in each study. Interestingly, this study established a new benchmark for pattern-based studies in terms of the number of rules and the performance achieved. Compared to studies that only employed pattern-based rules without any optimisation enhancements, this study attained superior precision and F-measure for explicit feature extraction. *Tubishat, Idris & Abushariah (2021)* also obtained the highest performance results with an optimiser. Nevertheless, *Mahmood, Abbas & ur Rehman (2023)*, *Tran, Duangsuwan & Wettayaprasit (2021)*, and *Pak & Günal (2022)* indicated lower outcomes.

Table 6 displays the comparative F-measure performance of the proposed algorithm concerning various benchmark studies for explicit feature extraction across multiple domain-specific datasets. The enhanced heuristic pattern-based algorithm consistently achieved the highest F-measure across all evaluated domains, surpassing other recent studies. Thus, substantial empirical evidence for the robustness and domain independence of the proposed method in explicit feature extraction tasks was effectively demonstrated.

## Performance evaluation of explicit feature extraction using enhanced heuristic pattern-based algorithm in multi-domain datasets

Table 7 presents the performance achieved by the enhanced heuristics pattern-based algorithm using multi-domain datasets. A comparative analysis concerning precision scores among datasets then indicated that all datasets attained a precision exceeding 90% (except for the SemEval 2016 Laptop dataset). On the contrary, the precision was only 1% below 90%. The recall for all datasets also exceeded 85%. This outcome implied that testing the proposed algorithm across multiple multi-domain datasets demonstrated consistent performance, which was characterised by stable and reliable precision, recall, and

**Table 5** Benchmarking enhanced heuristic pattern-based algorithm for explicit feature extraction on single domain (Amazon Product Review dataset).

| Studies | Precision | Recall | F-measure | Remarks |
|---|---|---|---|---|
| This study | 0.91 | 0.88 | 0.89 | 41 pattern rules |
| *Mahmood, Abbas & ur Rehman (2023)* | 85%, | 75% | 79% | Noun based extraction |
| *Pak & Günal (2022)* | NA | NA | 0.70 | Auto-generated rules |
| *Tran, Duangsuwan & Wettayaprasit (2021)* | 0.89 | 0.76 | 0.81 | 20 pattern-rules |
| *Tubishat, Idris & Abushariah (2021). (without the optimizer)* | 0.75 | 0.97 | 0.84 | 126 rules without the optimizer |
| *Tubishat, Idris & Abushariah (2021). (with optimizer)* | 0.92 | 0.93 | 0.92 | 57 rules with optimizer |
| *Chauhan & Meena (2020)* | 0.88 | 0.85 | 0.86 | Noun based extractions |
| *Rana & Cheah (2019)* | 0.86 | 0.91 | 0.89 | 10 pattern-rules |
| *Asghar et al. (2019)* | 0.83 | 0.71 | 0.77 | 10 pattern-rules |
| *Rana & Cheah (2017)* | 0.87 | 0.92 | 0.89 | 10 pattern-rules |
| *Kang & Zhou (2017)* | 0.87 | 0.88 | 0.87 | 7 dependency rules with 8 patterns |
| *Samha & Li (2016)* | 0.83 | 0.87 | 0.77 | 16 dependency rules |
| *Liu et al. (2016)* | 0.85 | 0.91 | 0.88 | 8 pattern-rules |
| *Khan & Jeong (2016)* | 0.81 | 0.82 | 0.8 | 9 pattern-rules |
| *Maharani, Widyantoro & Khodra (2015)* | 0.63 | 0.73 | 0.67 | 24 pattern-rules |
| *Khan, Baharudin & Khan (2014)* | 0.79 | 0.72 | 0.75 | 16 pattern-rules |
| *Htay & Lynn (2013)* | 0.73 | 0.86 | 0.79 | 8 pattern-rules |
| *Bagheri, Saraee & de Jong (2013)* | 0.86 | 0.64 | 0.73 | 4 pattern-rules |
| *Qiu et al. (2011)* | 0.88 | 0.83 | 0.86 | 4 pattern-rules |

F-measure indicators. Hence, the model was concluded to be generalisable and domain-independent.

Given that the precision values were consistently exceeding 90% across all datasets, this observation signified that the proposed algorithm effectively extracted a greater number of relevant features. Despite the recall for several datasets also being lower than 90%, the deviation was relatively minor (ranging from 2% to 4% below this threshold). This finding indicated that the proposed algorithm adequately extracted a greater number of relevant features compared to irrelevant ones. Overall, the evaluation results indicated that the proposed algorithm could improve the feature extraction performance.

This study applied the one-tailed paired $t$-test, assessing whether a statistically significant difference existed between the means of bench studies and the enhanced heuristic pattern-based algorithm. The tested null ($H_0$) and alternative hypotheses ($H_1$) are as follows:

i) $H_0$: No significant difference in performance is observed between the enhanced heuristic pattern-based algorithm and baseline methods across various domains.

ii) $H_1$: The enhanced heuristic pattern-based algorithm yields superior performance compared to baseline methods across various domains.

The F-measure results from the evaluation runs of six multi-domains (see Tables 5 and 6) were utilised in comparison to the benchmark results. Consequently, a $t$-statistic of

**Table 6 Benchmarking enhanced heuristic pattern-based algorithm for explicit feature extraction on multi-domain datasets.**

| Dataset | Studies | Techniques | F-measure |
|---|---|---|---|
| MAMS 2019 (Restaurant dataset) | This study | Pattern-based algorithm | 0.94 |
| | *Chen et al. (2022a)* | Discrete opinion tree GCN-using BERT | 0.84 |
| | *Tian, Chen & Song (2021)* | Type-Aware GCN using BERT | 0.83 |
| SemEval 2016 (Restaurant dataset) | This study | Pattern-based algorithm | 0.92 |
| | *Kabir, Othman & Yaakub (2024)* | Frequency-based syntax dependency with CRF | 0.91 |
| | *Zhao, Zhou & Xu (2024)* | Syntax and semantics with deep learning | 0.75 |
| | *Suwanpipob, Arch-Int & Wunnasri (2024)* | Linked open data (LOD) +Ngram | 0.77 |
| | *Venugopalan & Gupta (2022)* | LDA + Bert | 0.75 |
| | *Chen et al. (2022b)* | Frequency with deep learning | 0.80 |
| SemEval 2016 (Laptop dataset) | This study | Pattern-based algorithm | 0.88 |
| | *Busst et al. (2024)* | Ensemble BiLSTM | 0.54 |
| SemEval 2014 (Restaurant dataset) | This study | Pattern-based algorithm | 0.92 |
| | *Suwanpipob, Arch-Int & Wunnasri (2024)* | Linked open data (LOD)+Ngram | 0.80 |
| | *Kaur & Sharma (2023)* | HFV+LTSM | 0.92 |
| | *Li, Li & Xiao (2023)* | APSCL-BERT (Supervised deep learning) | 0.81 |
| | *Venugopalan & Gupta (2022)* | LDA + Bert | 0.81 |
| | *Pak & Günal (2022)* | Sequential pattern-based rule mining | 0.64 |
| | *Bie & Yang (2021)* | MTMVN | 0.79 |
| | *Zhu et al. (2020)* | SentiVec | 0.86 |
| | *Ayub et al. (2023)* | Ngram+TF-IDF+SVM | 0.85 |
| | *Li, Chow & Zhang (2020)* | SEML | 0.83 |
| SemEval 2014 (Laptop dataset) | This study | Pattern-based Algorithm | 0.89 |
| | *Li, Li & Xiao (2023)* | APSCL-BERT (Supervised deep learning) | 0.78 |
| | *Haq et al. (2023)* | LDA &BERTopic Modeling (Semi-supervised) | 0.72 |
| | *Pak & Günal (2022)* | Sequential pattern-based rule mining | 0.59 |
| | *Chen et al. (2022b)* | Frequency with deep learning | 0.70 |

**Table 7 Explicit feature extraction using enhanced heuristics pattern-based algorithm on multi-domain datasets.**

| No | Dataset | Precision | Recall | F-measure |
|---|---|---|---|---|
| 1 | Amazon Product Review (2004) | 0.91 | 0.88 | 0.89 |
| 2 | *Toledo-Ronen et al. (2022)* | 0.95 | 0.91 | 0.93 |
| 3 | MAMS (2019) | 0.96 | 0.93 | 0.94 |
| 4 | Sem Eval 2016-Restaurant | 0.94 | 0.89 | 0.92 |
| 5 | Sem Eval 2016-Laptop | 0.89 | 0.86 | 0.88 |
| 6 | Sem Eval 2014-Restaurant | 0.93 | 0.92 | 0.92 |
| 7 | Sem Eval 2014-Laptop | 0.90 | 0.88 | 0.89 |
| | **Average** | 0.93 | 0.90 | 0.91 |

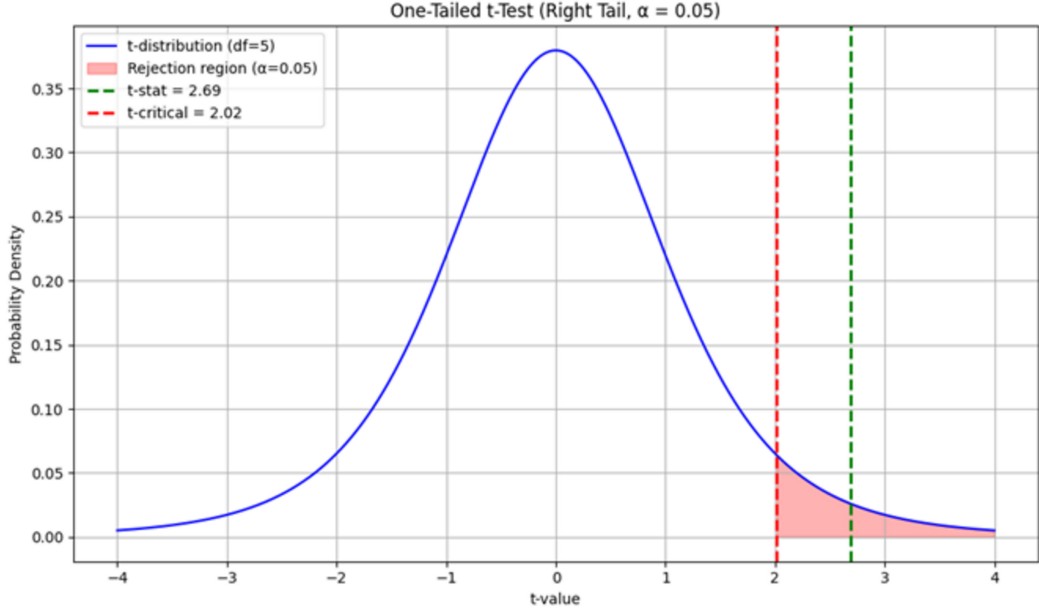

**Figure 8 One-tailed paired T-test for enhanced heuristic pattern-based algorithm performance.**

2.69 was computed, surpassing the critical $t$-value of 2.015 at a 95% confidence level. This outcome also produced a $p$-value of 0.0216 (<0.05). Figure 8 illustrates the results of the test, in which the $H_0$ was rejected. This finding suggested that the performance of the enhanced heuristic pattern-based algorithm was statistically significant, supporting $H_1$ (algorithm outperformed the benchmark studies significantly.

## CONCLUSION

The Kano model posits that businesses should transform their strategies by identifying customer needs, determining essential product features to address those needs, and rectifying product deficiencies. Considering that automation frequently neglects nuanced aspects, an enhanced extraction methodology is pivotal. This issue has prompted numerous researchers to explore methods for identifying a wide range of product features in reviews. Nevertheless, previous studies utilising pattern rules for feature extraction indicated that no singular process could identify all features within a review document. Although explicit feature extraction is less complex than feature extraction (requiring deeper processing), the patterns of extraction remain inadequately investigated. Therefore, the absence of relevant features in feature extraction presents a challenge that necessitates enhancements in existing methodologies.

This study successfully explored initiatives aimed at enhancing explicit feature extraction performance by identifying rules that could extract features overlooked by previous patterns. The analysis also created an enhanced heuristic pattern-based algorithm that utilised these rules for superior performance. Therefore, 16 newly constructed rules were formed to accomplish this objective. This study then presented a combination of 41 enhanced pattern rules, integrating newly established rules with 25 rules from previous

studies. As a result, an average precision of 0.93, a recall of 0.90, and an F-measure of 0.91 for seven datasets across multiple domains were observed, outperforming state-of-the-art studies. This performance achievement suggested that a greater number of relevant features were being accurately identified.

This study effectively achieved its objective of enhancing explicit feature extraction performance in customer reviews. Nonetheless, future studies should extend the algorithm to evaluate implicit and comparative feature extractions. Even though extracting implicit features and complex sentiments has not been currently implemented, it will be considered for future extension of this study due to its promising potential. This algorithm should also be enhanced to address negations and sarcasm. Moreover, future studies should readily expand the algorithm to encompass short sentence processing due to its text chunk-based operations. One notable example is opinion mining on Twitter, with little to no modifications required.

### Funding
The authors received no funding for this work.

### Competing Interests
The authors declare that they have no competing interests.

### Author Contributions
- Rajeswary Santhiran conceived and designed the experiments, performed the experiments, analyzed the data, performed the computation work, prepared figures and/or tables, and approved the final draft.
- Kasturi Dewi Varathan conceived and designed the experiments, analyzed the data, authored or reviewed drafts of the article, and approved the final draft.
- Yin Kia Chiam conceived and designed the experiments, analyzed the data, authored or reviewed drafts of the article, and approved the final draft.

### Data Availability
The code is available at GitHub and Zenodo:

- https://github.com/rajeswary3/feature-extraction.

- rajeswary3. (2023). rajeswary3/feature-extraction: Feature Extraction in Opinion Mining v1.0 (FeatureExtraction). Zenodo. https://doi.org/10.5281/zenodo.8365767.

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
