# Peer review of "Advancing opinion mining with optimised explicit feature extraction in customer reviews"

_PeerJ Computer Science, doi:10.7717/peerj-cs.3317_

## Round 0.1 · original submission · Major Revisions

· Academic Editor

Major Revisions

**Language Note:** PeerJ staff have identified that the English language needs to be improved. When you prepare your next revision, please either (i) have a colleague who is proficient in English and familiar with the subject matter review your manuscript, or (ii) contact a professional editing service to review your manuscript. PeerJ can provide language editing services - you can contact us at [email protected] for pricing (be sure to provide your manuscript number and title). – PeerJ Staff

Reviewer 1 ·

Basic reporting

The paper has clearly explained the gap in feature extraction from customer reviews and has addressed it in a step-by-step way with good coverage of existing methods.

However, the paper can be improved by considering the following suggestions:

1. Add comparison with recent deep learning models.
2. Include statistical significance testing.
3. Explain the pruning strategies more clearly with theoretical support.
4. Discuss how the method can handle implicit features or complex sentiments.

Experimental design

The following points are suggested for improving the experimental design:

1. Although the paper reports improved performance metrics, it does not include statistical significance testing. It is recommended to perform statistical tests to validate the improvements.
2. Include confidence intervals or p-values to support the reported performance results.
3. It would strengthen the paper if experimental graphs or case studies are provided to show how the pruning strategies impact the final feature set size and extraction accuracy.

Validity of the findings

The paper has verified performance only through standard metrics (precision, recall, F1-score) but does not validate against modern baseline methods such as transformer-based models. It would be better to include comparisons against recent deep learning models to ensure the validity of the proposed approach in current research contexts.

Reviewer 2 ·

Basic reporting

Formal results should include clear definitions of all terms and theorems, and detailed proofs.

Experimental design

No comment

Validity of the findings

No comments

Additional comments

The paper is well structured but following observations are there-

Pre-processing phase should be elaborated by explaining about the tools used.
The working of "Enhanced Heuristics Pattern-based Algorithm" should be explained in more detail for better understanding.
The details of "Opinion Lexicon" is missing. The authors should mention, that it is self created or being downloaded.

---

## Round 0.2 · accepted · Accept

· Academic Editor

Accept

The paper may be accepted.

Reviewer 1 ·

Basic reporting

The paper is well-structured with logical flow, and the authors have addressed the concerns.
The language has improved, but can still be simplified in some long sentences.
The figures and tables are presented good, but the captions should explain more.
The authors have fulfilled my concerns in this section.

Experimental design

In this revision, the authors have improved the following:
1. The research question is well defined.
2. The methodology is explained with good detail and described with examples.
3. Datasets are discussed clearly.
4. The comparative analysis with baseline studies is added.

Validity of the findings

In the revised version, the results are statistically sound and consistent across multiple datasets. The t-test validation is properly explained. The limitations and future work are now included.

Additional comments

The revised version is improved and suitable for publication. The authors have addressed and responded to the concerns raised in the previous version.